# Improving Weakly Supervised Lesion Segmentation using Multi-Task Learning

**Tianshu Chu***                                                    TC2992@NYU.EDU
*New York University*

**Xinmeng Li***                                                     XL1575@NYU.EDU
*New York University*

**Huy V. Vo**                                                VAN-HUY.VO@INRIA.FR
*Ecole Normale Superieure, INRIA and Valeo.ai*

**Ronald M. Summers**                                     RSUMMERS@CC.NIH.GOV
*National Institutes of Health Clinical Center*

**Elena Sizikova**                                                  ES5223@NYU.EDU
*New York University*
*\* - contributed equally*

## Abstract

We introduce the concept of multi-task learning to weakly-supervised lesion segmentation, one of the most critical and challenging tasks in medical imaging. Due to the lesions' heterogeneous nature, it is difficult for machine learning models to capture the corresponding variability. We propose to jointly train a lesion segmentation model and a lesion classifier in a multi-task learning fashion, where the supervision of the latter is obtained by clustering the RECIST measurements of the lesions. We evaluate our approach specifically on liver lesion segmentation and more generally on lesion segmentation in computed tomography (CT), as well as segmentation of skin lesions from dermatoscopic images. We show that the proposed joint training improves the quality of the lesion segmentation by 4% percent according to the Dice coefficient and 6% according to averaged Hausdorff distance (AVD), while reducing the training time required by up to 75%.

**Keywords:** Lesion Localization, Weakly Supervised Learning, Multi-Task Learning

## 1. Introduction

Assessing lesion and tumor growth is a central problem in medical imaging for oncology, required for assessing cancer burden and aiding radiologists in accurately labeling important findings. To analyze lesions, radiologists typically manually annotate computed tomography (CT) scans containing lesions with response evaluation criteria in solid tumors (RECIST) measurements, consisting of the major and minor axes of the best-fit ellipse that coarsely describes the lesion segmentation mask (Eisenhauer et al., 2009). This process, however, does not provide accurate pixel-level segmentation that would be used to monitor lesion shape, and may prove crucial to identifying any abnormalities.

In this work, we propose a new automatic method for predicting lesion segmentations that leverages existing RECIST annotations (Yan et al., 2018) as training data in a weakly-

supervised learning setting. In particular, we show that jointly segmenting and classifying lesions according to their shapes in a multi-task learning fashion achieves better segmentation accuracy than learning a segmentation network alone. This insight is motivated by the successful application of multi-task learning in other domains and problems (Zhang and Yang, 2017; Standley et al., 2020), and the idea that clustering provides a strong supervision signal for training computational models (Caron et al., 2018). Unlike other recent weakly supervised lesion segmentation methods (Agarwal et al., 2020a,b), we do not use co-segmentation, and instead allow the model to learn appropriate lesion similarities via clustering, thereby improving segmentation quality and reducing training time. We generate class labels by clustering lesions using their RECIST measurements and learn a model to simultaneously classify and segment input CT slices with lesions. We compare our approach to the state-of-the art medical imaging weakly supervised baseline (Agarwal et al., 2020a) on CT and skin lesion segmentation and show that our method generates more accurate segmentations while requires significantly less training time.

Our contributions can be summarized as follows. We propose a new joint classification and localization scheme for training weakly-supervised lesion segmentation models. We demonstrate how to modify existing segmentation architectures to incorporate this new algorithm. We conduct a systematic analysis of the utility of the proposed new methodology and show that it quantitatively and qualitatively outperforms prior baselines in this domain.

## 2. Related Work

Manually producing dense pixel-wise segmentations for medical images is a time-consuming task that requires domain expertise. Hence, large-scale datasets required for training automatic segmentation models are not available for many medical imaging tasks (Guo et al., 2018; Tajbakhsh et al., 2020). On the other hand, weakly-supervised methods allow for the use of class labels (Hu et al., 2020) or coarse segmentation masks (Agarwal et al., 2020a; Xie et al., 2020). In particular, response evaluation criteria in solid tumors (RECIST) annotations (Eisenhauer et al., 2009) can be used in lieu of pixel-wise segmentation for training automatic segmentation models. RECIST measurements are often stored in hospital picture archiving and communication systems (PACS) and are annotated manually by radiologists during screening procedures (Cai et al., 2018). Annotations in the form of clicks (Roth et al., 2020) or bounding boxes (Rajchl et al., 2016) are also popular. Finally, weakly-supervised learning is often combined with interactive techniques to speed up image annotation during organ screening (Roth et al., 2020).

Machine learning models trained on a single task can ignore useful domain information that is contained in the training signals of other related, but different tasks, which hinders their generalization ability and performance. Multi-task learning (MTL) is an approach to alleviate this limitation (Caruana, 1998). MTL jointly trains multiple relevant tasks, usually with a shared presentation, to simultaneously improve their performance. Recently, MTL has been proven effective in multiple machine learning applications: Natural language processing (Liu et al., 2019), speech recognition (Deng et al., 2013), stock prediction (Zhou and Voigt, 2020), computer vision (Zhou et al., 2020) and medical image analysis (Le et al., 2019). In particular, MTL can be used to improve the performance of a single task by jointly training it with other weaker tasks whose training supervision can be obtained from

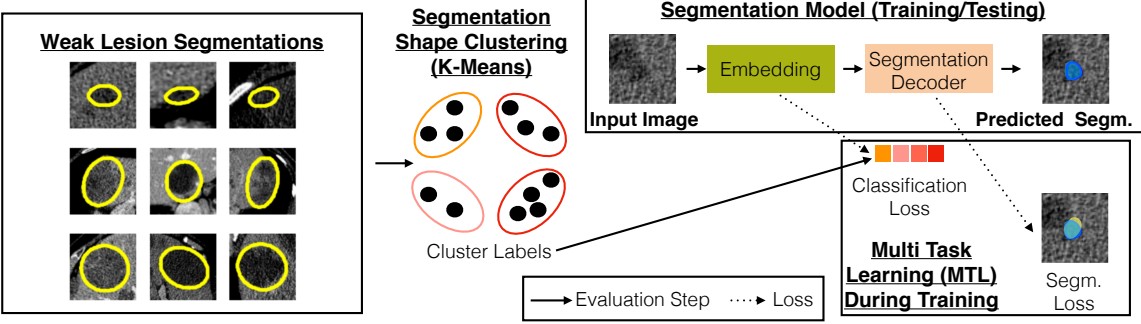

Figure 1: Overview of the proposed joint classification localization network. We first extract class labels by clustering the lesion RECIST shapes into $K$ classes (shown with $K = 4$). We then train a joint segmentation and classification model using the extracted labels.

the supervision of the main task (Girshick, 2015; Lee et al., 2019). Our work belongs to this particular branch of MTL. In medical imaging, MTL has also been employed for lesion segmentation (Yang et al., 2017) or analyzing COVID-19 CT images (Amyar et al., 2020) but these work require exact class labels for all training images and dense segmentations for at least a part of them. On the other hand, our work use only coarse segmentation supervision (RECIST measurements), and extracts class labels in an unsupervised manner.

## 3. Multi-task Learning Approach for Weakly-Supervised Segmentation

We now describe our multi-task model for weakly supervised lesion segmentation. An overview of our proposed method can be seen in Figure 1.

**Weakly-Supervised Lesion Segmentation using RECIST.** Let $R = (\{I_i, r_i\}|i = \{1, 2, \ldots, N\})$ be the set of images $I_i$ and their corresponding RECIST measures $r_i$ with $r_i = (x_i, y_i, w_i, h_i, a_i)$, where $(x_i, y_i)$ are the coordinates of the center, $w_i$ and $h_i$ are the width and height, and $a_i$ is the rotation angle of the RECIST ellipse[1]. We train a model, denoted by a function $f$, such that for a given image $I_i \in \mathbb{R}^{H \times W \times C}$ ($H$, $W$ and $C$ are the image's height, width and number of channels respectively), $f(I_i)$ is the lesion segmentation map of $I_i$, i.e., $f(I_i) \in \{0, 1\}^{H \times W}$ where $f(I_i)[p, q] = 1$ if pixel $(p, q)$ is contained in the lesion. The model consists of an encoder $E$ which produces an embedded representation of the input, and a decoder $D$ which transforms the embedded representation into a binary map, i.e., $f(I_i) = D(E(I_i))$. In the presence of a ground-truth segmentation map $s_i^{gt}$, $f$ is trained to minimize $\mathcal{L}^{seg}(f(I_i), s_i^{gt})$, the difference between the decoder's output and $s_i^{gt}$. Since ground-truth segmentation maps are not available in our setting, we instead generate pseudo-ground truth segmentation maps $s_i$ from RECIST measures and minimize $\mathcal{L}^{seg}(f(I_i), s_i)$, the difference between the decoder's output and $s_i$. We define $\mathcal{L}^{seg}$ to be the

---

1. Typically, RECIST is encoded via user clicks $\hat{R} = (x_{11}, y_{11}, x_{12}, y_{12}, x_{21}, y_{21}, x_{22}, y_{22})$ where $(x_j, y_j)$ denote the endpoints of the ellipse axis. To better capture characteristic features, we fit an ellipse to these measurements and use the transformed parameterization.

binary cross entropy (BCE) loss:

$$\mathcal{L}^{seg}(f(I_i), s_i) = -\frac{1}{HW} \sum_{p,q} s_i[p,q] \log f(I_i)[p,q] + (1 - s_i[p,q]) \log(1 - f(I_i)[p,q]) \tag{1}$$

where $s_i[p,q]$ and $f(I_i)[p,q]$ are respectively the values of $s_i$ and $f(I_i)$ at pixel $(p,q)$. Intuitively, by projecting the input image into a small embedding space with the encoder $E$, we force the model to learn only relevant features that are useful in reconstructing the pseudo-ground truth segmentation map $s_i$. The information learned in the embedded representation is therefore critical for the model performance. We propose to regularize this representation by training a lesion classification model simultaneously with the segmentation loss.

**Lesion Classification.** Given the class $c_{I_i} \in \{1, ..., K\}$ of image $I_i$, we train a classifier $g$ on the encoder's output with the cross-entropy loss:

$$\mathcal{L}^{cls}(g_{I_i}, c_{I_i}) = -\log\left(\frac{\exp(g_{I_i}[c_{I_i}])}{\sum_j \exp(g_{I_i}[j])}\right), \tag{2}$$

where $g_{I_i} = g(E(I_i))$ is the output of the classifier $g$ on image $I_i$, and $g_{I_i}[k]$ is the value of $g_{I_i}$ at index $k$. In our context, $c_{I_i}$ is generated from the RECIST measure as follows. The set of segmentation examples $S = \{s_1, s_2, \ldots, s_t\}$ is partitioned into $K \leq |S|$ sets $Q = \{q_1, q_2, \ldots, q_K\}$ (i.e., $K$ clusters) via the K-means algorithm (MacQueen et al., 1967). Subsequently, for every example $I_i$, where $i = 1, 2, \ldots N$, we use the index $c_{I_i}$ of its assigned cluster as the label for classification loss.

**Multi-task Learning.** We train our model in a multi-task learning fashion to minimize the joint loss:

$$\mathcal{L}(I_i, r_i) = \alpha_{seg}\mathcal{L}^{seg}(f(I_i), s_i) + \alpha_{cls}\mathcal{L}^{cls}(g_{I_i}, c_i), \tag{3}$$

where $\alpha_{seg}$ and $\alpha_{cls}$ are the segmentation and the classification loss weights, respectively. Since the segmentation and classification modules share the same encoder, useful information retained by the lesion classifier is transferred to the segmentation head for better performance. We now turn to experimental validation of the proposed learning framework.

## 4. Datasets, Metrics and Implementation Details

**Computed Tomography (CT) Images.** For evaluation on CT, we consider two publicly available datasets. The first dataset is the DeepLesion dataset (Yan et al., 2018) which consists of 32,735 computed tomography (CT) images with lung nodules, liver tumors, enlarged lymph nodes and other internal organ abnormalities from an anonymized set of 4,400 unique patients. Each lesion image is annotated with long and short diameter RECIST measurements. Since ground-truth pixel-level segmentations for this dataset are not available, we rely on approximate GrabCut-based segmentations as ground truth. We use the official data split for this dataset. The second dataset is LiTS (Bilic et al., 2019) which consists of CT images containing liver tumors from the Liver Tumor Segmentation Benchmark. This dataset has manually annotated dense segmentation masks for tumors in each CT slice but RECIST measure is not available. We therefore fit ellipses to these masks to obtain RECIST parameters. We use 11,522 images (48 volumes) for training, 3310 images (44 volumes) for validation and 1630 images (20 volumes) for testing.

**Dermatoscopic Images.** We evaluate the generalization of our method on dermatoscopic images of skin lesions. While RECIST is more commonly used for CT datasets, we use skin lesion as an additional test case due to the wider availability of publicly accessible data-sets and dense segmentation. We consider the HAM10K dataset (Tschandl et al., 2018), which consists of 10,015 dermatoscopic images and pixel-level segmentation masks of common pigmented skin lesions, such as carcinomas, keratoses and melanomas. Lesions in this dataset often have blurry boundaries and represent a challenging test case for our method. We split the dataset into 70%/20%/10% for training, validation, and testing subsets, respectively.

**Evaluation Metrics.** We use several standard image processing and medical imaging evaluation metrics to evaluate segmentation performance: Intersection over union (IoU), Dice coefficient, volumetric similarity and averaged Hausdorff distance. We also evaluate whether the segmentation faithfully captures the shape of the lesion with the center error and the perimeter error. The center error measures the Euclidean distance between the predicted and ground truth segmentation center. The perimeter error is the difference in length between the perimeter of the ground-truth segmentation and the model output. We define all metrics in the appendix. In particular, IoU, Dice, and VS measure volumetric simiarity, while AVD, center error and perimeter error evaluate the surface similarity.

**Model Architecture and Training Details.** We use the DeepLabV3+ model (Chen et al., 2018) with a ResNet101 (He et al., 2015) backbone as our encoder-decoder architecture, to which we refer as A1. This architecture relies on atrous spatial pyramid pooling (ASPP) to encode context at multiple scales and a decoder that refines segmentation boundaries. In the decoder part, the encoder features are first bilinearly upsampled by 4 and then concatenated with the corresponding low-level features after applying the 1x1 convolution. After the concatenation, two 3x3 convolutions with 256 filters are used to refine the features followed by another simple bilinear up-sampling by the factor of 4. This architecture has been widely popular for semantic segmentation in both non-medical (Lateef and Ruichek, 2019) and medical (Anderson et al., 2021; Khan et al., 2020) imaging semantic segmentation tasks. We further modify this architecture to add a classification head with $K$ class outputs at the bottleneck layer to test the effect of added class labels, to which we refer as A1+L. Code for our experiments is made publicly available [2].

Similar to prior work (Agarwal et al., 2020a,b), we focus on the segmentation problem only, and assume that the lesion location has already been given by RECIST measurements. Our input data therefore consists of patches around each lesion. In DeepLesion, we use the center of the provided bounding box and crop a square patch of side length 120 pixels. For LiTS, we estimate the bounding box with paddings of 20 pixels. The HAM10K dataset is already provided in a patch form and required no pre-processing. To compare with prior results, we evaluate the co-segmentation method of Agarwal et al. (2020a), to which we refer as ACoseg. We implement this approach using the same encoder and decoder as A1, but add two branches and co-attention in order to perform co-segmentation. Please see Agarwal et al. (2020a) for more details. We train both models for 50 epochs with early stopping by segmentation loss on the validation set. We use Adam optimizer with learning rate 0.001 and batch size 10 for A1 and A1+L and learning rate $10^{-5}$ with batch size 8 for ACoseg.

---

2. Link to code repository: https://github.com/esizikova/weaklesionmtl

We use $\alpha_{seg} = 50$ and $\alpha_{cls} = 1.0$ in all of the multi-task experiments. Finally, we perform an ablation study (see Figure 2) on the number of clusters, and chose $K_{opt} = 190$ for NIH DeepLesion, $K_{opt} = 45$ for HAM10K, and $K_{opt} = 40$ for LiTS. We report additional ablation studies and a comparison to UNet (Ronneberger et al., 2015) in supplementary material.

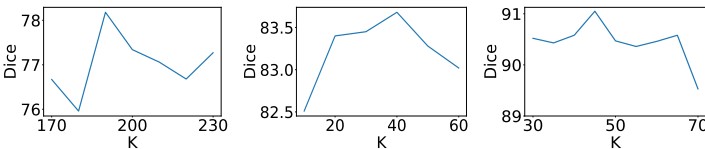

Figure 2: An illustration of the segmentation quality (Dice coefficient) as a function of the number of lesion clusters ($K$) for the DeepLesion (left), LiTS (middle) and HAM10K (right) datasets.

## 5. Experimental Results

We now quantitatively and qualitatively evaluate our multi-task learning scheme for weakly supervised lesion segmentation.

**Visualization of Lesion Clusters.** We visualize the groups of lesions obtained by K-means clustering on each dataset in Figure 3. While there exists significant variation of lesion attributes such as size, shape and intensity within all datasets, the clustering procedure groups lesions into classes of similar size, position, and relative rotation. Cluster assignment is therefore a reasonable supervision for lesion classification.

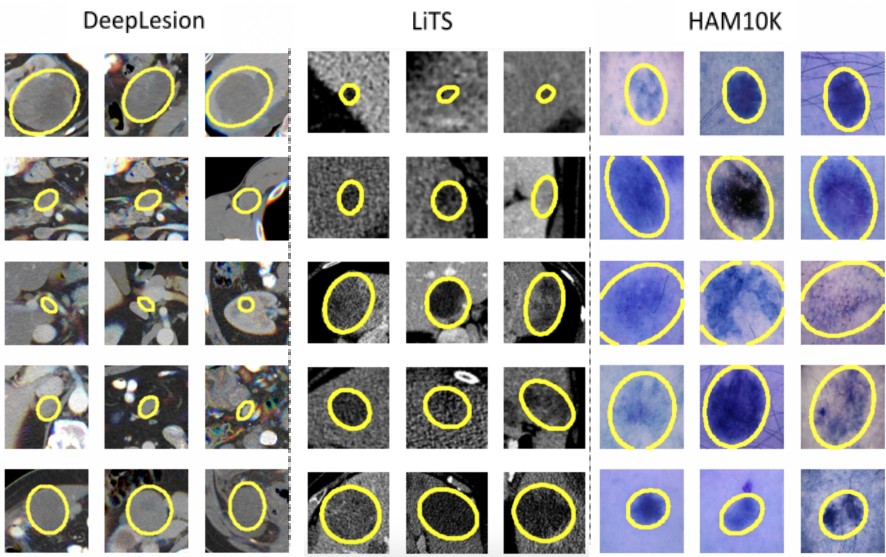

Figure 3: Visualization of lesion groups obtained by clustering ellipse measurements. In a dataset, each row corresponds to a cluster. Clustering groups similar examples within each of the considered CT (DeepLesion, LiTS) and dermatoscopic (HAM10K) datasets.

| Dataset | Model | IoU | Dice | VS | Center Error | Perimeter Error | AVD |
|---|---|---|---|---|---|---|---|
| NIH DeepLesion | A1 | 63.02±1.47 | 75.36±1.25 | 80.65±1.60 | 3.19±0.06 | 9.21±0.64 | 7.37±0.24 |
| | ACoseg | 62.75±0.42 (-0.43%) | 74.38±0.77 (-1.30%) | 80.58±0.49 (-0.09%) | 5.56±1.19 (+74.29%) | 8.43±0.3 (-8.47%) | 11.84±2.24 (+60.65%) |
| | A1+L | **65.53±0.71 (+3.98%)** | **77.43±0.57 (+2.75%)** | **83.31±0.70 (+3.30%)** | **3.09±0.02 (-3.13%)** | **7.79±0.38 (-15.42%)** | **6.92±0.19 (-6.11%)** |
| LiTS | A1 | 71.44±0.68 | 82.68±0.51 | 89.07±0.64 | 2.00±0.03 | 9.58±0.24 | 9.25±0.17 |
| | ACoseg | 71.86±0.34 (+0.59%) | 82.92±0.26 (+0.29%) | 88.81±0.22 (-0.29%) | 1.90±0.06 (-5.00%) | 8.99±0.16 (-6.16%) | **8.90 ± 0.07 (-3.78%)** |
| | A1+L | **72.76±0.14 (+1.85%)** | **83.62±0.09 (+1.14%)** | **89.94±0.16 (+0.98%)** | 1.96±0.09 (-2.00%) | **8.27±0.05 (-13.67%)** | 9.05±0.20 (-2.16%) |
| HAM10K | A1 | 82.65±0.12 | 90.01±0.09 | 91.84±0.03 | 2.57±0.09 | 9.20±0.26 | 10.23±0.17 |
| | ACoseg | 83.19±0.86 (+0.65%) | 90.42±0.51 (+0.46%) | 92.10±0.77 (+0.28%) | 2.42±0.03 (-5.84%) | 9.77±0.59 (+6.20%) | **9.61±0.31 (-6.06%)** |
| | A1+L | **83.51±0.26 (+1.04%)** | **90.54±0.22 (+0.59%)** | **92.82±0.34 (+1.07%)** | 2.52±0.09 (-1.95%) | **8.87±0.4 (-3.59%)** | 9.73±0.14 (-4.89%) |

Table 1: Evaluation of the effect of multi-task learning on segmentation quality using A1 and ACoseg architectures. Best result is shown in bold. For IoU, Dice, and VS metrics, higher number is better. For Center Error, Perimeter Error, and AVD, lower number is better.

**Quantitative Segmentation Performance.** We investigate the effectiveness of our joint classification and segmentation training for lesion segmentation in Table 1. The classification+segmentation multi-task training A1+L outperforms the single-task training A1 on all datasets and according to all metrics. In particular, the Dice coefficient is improved by about $1 - 3\%$ compared to the non-classification baseline A1 on all datasets, and by $1 - 4\%$ compared to the co-segmentation ACoseg baseline on the CT datasets. The proposed method also significantly improves the surface and center-based metrics: Our method generates $2\% - 6\%$ better segmentations with respect to AVD, improves the center estimation by up to 3%, and perimeter estimation by up to 15%. On HAM10K and LiTS, ACoseg slightly outperforms or matches our A1+L model according to AVD. However, we observe that it exhibits more segmentation artifacts: In some cases, it cannot generate a mask and outputs a zero prediction. For evaluation purposes, in cases where a zero mask is predicted, the worst value of the same metric among other test cases is used. It is also noteworthy that ACoseg performs better than A1, validating the conclusion in (Agarwal et al., 2020b). Sample qualitative results of this experiment are reported in Figure 4.

**Training Time.** Adding label classification using our proposed multi-task learning setup offers not only quantitative segmentation improvement, but also a significant savings in training time. Table 2 shows the GPU time required to train the proposed model A1+L and the ACoseg baseline on each dataset. A1+L saves 69%-75% of training time (29.2 hours less on DeepLesion) compared to ACoseg, while achieving substantially better quantitative performance (see Table 1). The improvement can be seen in both smaller (LiTS, HAM10K) and larger (NIH DeepLesion) datasets. Note that the cluster classification head contains very few parameters compared to the backbone, and therefore the A1 and A1+L models require approximately the same amount of training time.

| Dataset | ACoseg | A1+L |
|---|---|---|
| NIH DeepLesion | 41.5 | 12.3 (-70%) |
| LiTS | 16.6 | 4.2 (-75%) |
| HAM10K | 12.5 | 3.9 (-69%) |

Table 2: Training time (GPU hours) until convergence of different models on the three datasets.

## 6. Conclusion

We have shown that generating labels from existing weak supervision and jointly train to segment and cluster lesions in a multi-task learning fashion allows more accurate segmen-

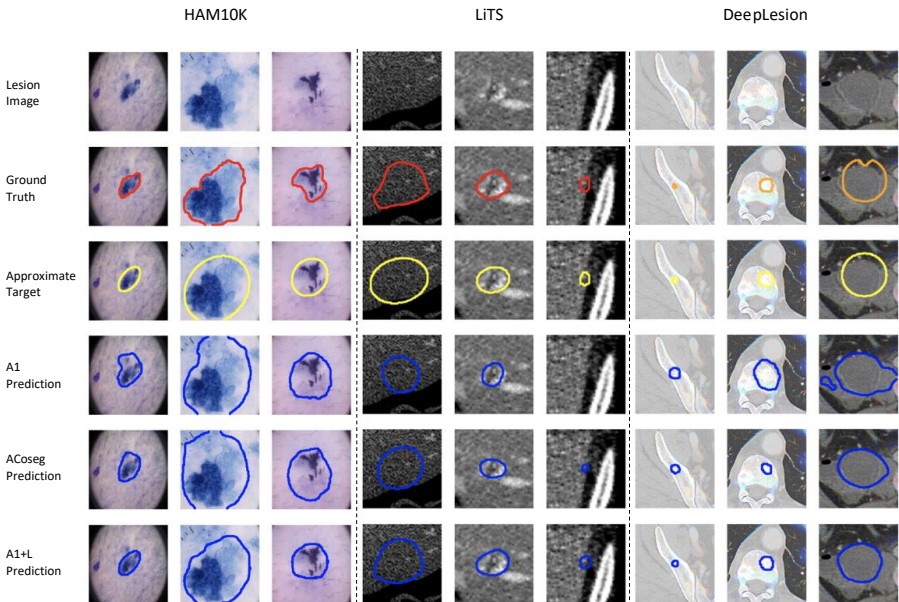

Figure 4: Visualization of the effect of clustering (denotes "+L") on segmentation prediction across datasets. For DeepLesion dataset, we use GrabCut to obtain segmentations because the ground truth is not available.

tations of computed tomography (CT) and dermatoscopic imaging data while significantly reducing training time. Our approach is of course only one of many potential ways to incorporate unsupervised learning into segmentation problems in medical imaging tasks. As unsupervised and weakly-supervised learning is only starting to become more widely used in medical imaging, there are a number of future directions this approach could be extended. In particular, it would be useful to understand the performance of the proposed methodology on other types of medical images, such as chest radiographs (CXR), magnetic resonance imaging (MRI) and others. Also, it would be important to investigate how other types of data typically present in radiology reports can be used for a stronger supervision signal. We hope that our exploration into weakly-supervised MTL learning for lesion segmentation will encourage more research applications in this domain.

## 7. Acknowledgements

TC, XL, ES were supported by the Moore-Sloan Data Science Environment initiative (funded by the Alfred P. Sloan Foundation and the Gordon and Betty Moore Foundation) through the NYU Center for Data Science. HVV was supported in part by a Valeo/Prairie CIFRE PhD Fellowship. RMS was supported by the Intramural Research Program of the National Institutes of Health Clinical Center.

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

## Appendix

In this section, we report additional results and ablation studies using the studied models.

### 7.1. Metric Definition

Let $\mathcal{A}^t$ be the pixels in the ground truth segmentation area, and $\mathcal{A}^p$ is be the pixels in the predicted segmentation area. Then the IoU metric is defined by: $M_{iou}(\mathcal{A}^p, \mathcal{A}^t) = |\mathcal{A}^t \cap \mathcal{A}^p|/|\mathcal{A}^t \cup \mathcal{A}^p|$. The Dice coefficient is defined as: $M_{dice}(\mathcal{A}^p, \mathcal{A}^t) = 2 \times |\mathcal{A}^t \cap \mathcal{A}^p|/(|\mathcal{A}^t| + |\mathcal{A}^p|)$. Following Agarwal et al. (2020a), we measure averaged Hausdorff distance (AVD) and volumetric similarity (VS) (please see (Taha and Hanbury, 2015) for details). AVD is defined by: $M_{AVD}(A, B) = \max(d(A, B), d(B, A))$ where $d(\cdot)$ is the directed average Hausdorff distance: $d(A, B) = 1/|A| \sum_{a \in A} \min_{b \in B} \|a - b\|$. The volumetric similarity is defined as: $M_{VS} = 1 - |FN - FP|/(2TP + FP + FN)$, where $FN$, $FP$, $TP$ are the false negative, false positive, and true positive cardinalities, respectively. The center error is defined as $E_{ctr} = \sqrt{(x_t - x_p)^2 + (y_t - y_p)^2}$ where $(x_t, y_t)$ are coordinates of the ground truth segmentation center and $(x_p, y_p)$ are coordinates of the predicted segmentation center (we calculate the leftmost, rightmost, top and bottom segmentation points and use them to calculate the center). The perimeter error is the difference in length between the perimeter of the ground-truth segmentation and the model output, mathematically defined as $E_{ptr} = |\mathcal{P}_t - \mathcal{P}_p|$ where $\mathcal{P}_t$ is the length (number of pixels) of the outermost edge of the ground truth segmentation, and $\mathcal{P}_p$ is the length of the predicted segmentation.

### 7.2. Effect of Supervision Type

In Tables 3, 4, and 5, we report ablation studies comparing the effect of adding classification to the network when using ellipse (RECIST) and GrabCut supervision on all of the considered datasets. It can be seen that adding labels helps improve results according to IoU, Dice and Center Error with all supervisions. On NIH DeepLesion (Yan et al., 2018), GrabCut-based method A1+L generates the best results according to IoU an Dice, but ellipse-based method A1+L generates a slightly more accurate lesion center. On LiTS (Bilic et al., 2019), the ellipse-based method A1+L significantly outperforms GrabCut, due to the difficulty of generating GrabCut for the lesions found in this dataset that have a dark background. On HAM10K (Tschandl et al., 2018), the ellipse-based methods generate best results according to all metrics, except circumference error.

| Method | IoU | Dice | Center Error | Circumf. Error |
|---|---|---|---|---|
| A1 on GrabCut | 65.55 | 76.82 | 3.84 | 8.06 |
| A1+L on GrabCut | 69.78 | 80.52 | 3.09 | 6.81 |
| A1 on Ellipse | 61.17 | 73.77 | 3.24 | 9.75 |
| A1+L on Ellipse | 66.49 | 78.18 | 3.06 | 7.34 |

Table 3: Evaluation of segmentation in the NIH DeepLesion (Yan et al., 2018) dataset using different input segmentations.

| Method | IoU | Dice | Center Error | Circumf. Error |
|---|---|---|---|---|
| A1 on GrabCut | 70.86 | 82.14 | 2.18 | 9.22 |
| A1+L on GrabCut | 72.41 | 83.30 | 2.18 | 8.56 |
| A1 on Ellipse | 70.89 | 82.27 | 2.00 | 9.64 |
| A1+L on Ellipse | 72.84 | 83.69 | 1.97 | 8.21 |

Table 4: Evaluation of segmentation quality in the LiTS (Bilic et al., 2019) dataset using different input segmentation.

| Method | IoU | Dice | Center Error | Circumf. Error |
|---|---|---|---|---|
| A1 on GrabCut | 83.30 | 90.39 | 2.73 | 8.79 |
| A1+L on GrabCut | 83.63 | 90.56 | 2.64 | 8.69 |
| A1 on Ellipse | 82.52 | 89.92 | 2.63 | 8.86 |
| A1+L on Ellipse | 83.87 | 90.85 | 2.50 | 9.38 |

Table 5: Evaluation of segmentation quality in the HAM10K (Tschandl et al., 2018) dataset using different input segmentation.

## 7.3. Comparison with Fully Supervised Methods

Next, we compare how results of the weakly-supervised methods compare with results of their fully-supervised counterparts. In Table 6, we compare models trained with ellipse-based (weak) segmentation, GrabCut (weak) supervision and dense, pixel-wise full segmentation supervision. For NIH DeepLesion (Yan et al., 2018) dataset, we use GrabCut-based segmentation as ground truth since human-annotated dense segmentation is not publicly available. While fully-supervised models achieve more accurate (according to Dice) segmentation, weak supervision reduces performance by about 3%, demonstrating the utility of such methods on lesion segmentation tasks. On LiTS, the ellipse based MTL segmentation method slightly outperforms the fully-supervised segmentation baseline. This is likely due to the fact that the ellipse prior helps regularize the model and prevent it from generating incorrectly-shaped outliers.

| Dataset | Model | | |
|---|---|---|---|
| | GrabCut | Ellipse | Fully-Supervised |
| NIH DeepLesion | 80.52* | 78.18 | 80.52* |
| LiTS | 83.30 | 83.62 | 82.87 |
| HAM10K | 90.56 | 90.85 | 93.53 |

Table 6: Evaluation of the effect of supervision type on segmentation quality. Results are reported according to the Dice metric. *-For DeepLesion we use GrabCut-based segmentation as ground truth as human-annotated dense segmentation is not publicly available.

## 7.4. Effect of a Different Backbone

In Table 7 we compare the segmentation quality when using the DeepLabV3+ (Chen et al., 2018) and the U-Net (Ronneberger et al., 2015) backbone networks. We find that DeepLabV3+ architecture slightly outperforms U-Net.

| Dataset | U-Net | DeepLabV3+ |
|---|---|---|
| NIH DeepLesion | 76.76 | **76.83** |
| LiTS | 83.28 | **83.40** |
| HAM10K | 89.86 | **89.97** |

Table 7: Performance comparison (Dice) between U-Net and DeepLabV3+ architectures.

### 7.5. Hyperparameter Selection and Classification Performance

In Figure 5, we perform an ablation study of the segmentation loss weight $\alpha_{seg}$ using Dice performance on the HAM10K dataset (Tschandl et al., 2018). We find that the performance gradually increases up to $\alpha_{seg} = 50$, after which it drops slightly. For low values of $\alpha_{seg}$, the classification loss likely significantly outweighs the segmentation loss, and performance drops. With $\alpha_{seg} = 50$, optimal performance is achieved. With larger values, the segmentation loss outweighs classification, and no longer helps performance. In Table 8, we additionally report the train and test classification a of the A1+L model across all datasets.

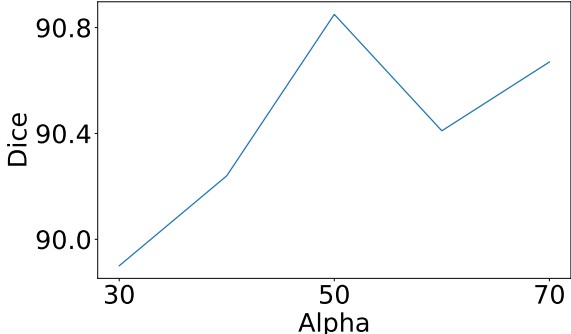

Figure 5: Ablation study of segmentation accuracy (Dice coefficient) as functions of segmentation loss weight $\alpha_{seg}$ for the HAM10K dataset (Tschandl et al., 2018) and A1+L model.

| Dataset | Train | Test |
|---|---|---|
| NIH DeepLesion | 16.9 | 12.1 |
| LiTS | 48.5 | 23.9 |
| HAM10K | 54.0 | 46.7 |

Table 8: Classification accuracy from A1+L model on different datasets.

### 7.6. Number of Model Parameters

We report the total number of parameters for each model. For A1 and ACoseg, the number of parameters is the same across all datasets: A1 requires 45,669,713 parameters and ACoseg requires 64,554,577 parameters. The number of parameters A1+L depends on $K$, the number of classes in the classification head: 46,059,023 for NIH DeepLesion, 45,751,673 for LiTS and 45,761,918 for HAM10K.

### 7.7. Additional Qualitative Results

In Figure 6, we report additional qualitative segmentation results. Our method appears to generate more accurate lesion boundaries with respect to the ground truth, as compared to both A1 and ACoseg baselines, introducing less border artifacts across all three considered datasets.

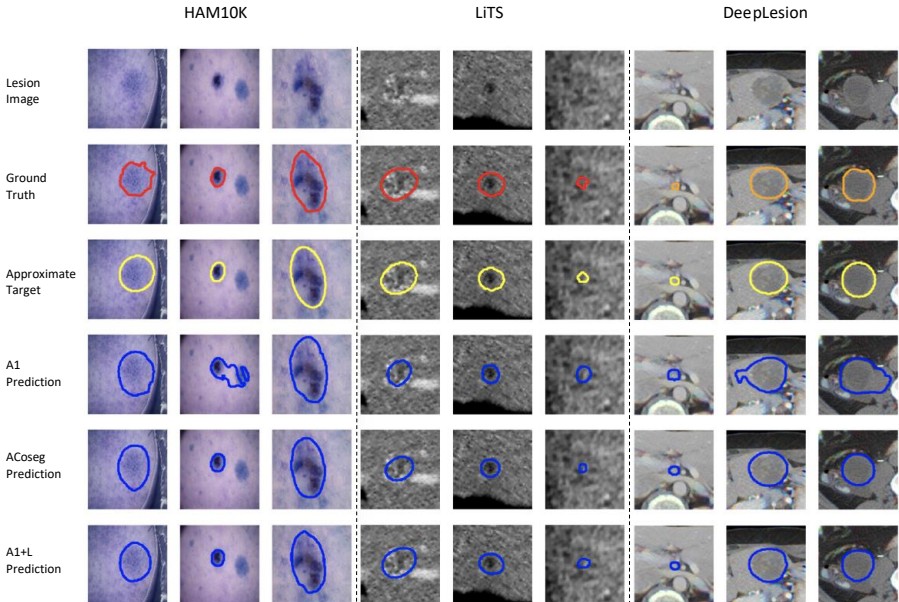

Figure 6: Visualization of the effect of clustering (denoted "+L") on segmentation prediction across datasets.

