# OpenReview forum: "Improving Weakly Supervised Lesion Segmentation using Multi-Task Learning"
_MIDL.io/2021/Conference — MIDL 2021_

### Official Review · AnonReviewer3 · 2021-03-05

**Confidence:** 4
**Preliminary Rating:** 2
**Final Rating:** 2

**Summary:**

This paper presents a multi-task learning-based method for lesion segmentation from CT images. Besides the original mask (or weakly mask) labels, they also use K-means to get the pseudo cluster label. They use this pseudo-label supervision as a second classification task for the network, aiming to improve the generalization capability of the network. The authors evaluate the proposed network on three networks and the proposed multi-task learning framework outperforms the baselines.

**Strengths:**

1. The key idea of multi-task learning is clear and reasonable.
2. The proposed method show improvements on three public lesion segmentation datasets. The authors show conduct ablation study to show how to choose the important hyperparameter k.

**Weaknesses:**

1. The technical novelty is limited, as the unsupervised task (classification task) adopts the k-means to generate the pesudo-labels. This cluster-based unsupervised learning has been widely investigated in the computer vision or medical imaging domain.

2. For me, the improvement on NIH deeplesion dataset (8.7%) is not very convincing. As the lesions have similar shapes and it is difficult for me to understand this simple cluster task would bring such large improvement.

**Deanonymize Review:**

no

**Detailed Comments:**

1. In Table 1, are the results from the single experiment run or the average results of several runs? Is there any performance fluctuation among different experiment runs for the same method? I suggest the authors report the average results of several runs to remove the randomness of each experiment run to better show the effectiveness of the proposed method.
2. A typo in Figure 1. "Shown with K=4"?

**Final Rating Justification:**

The authors have not submitted the rebuttal.

**Justification Of The Preliminary Rating:**

Although the whole paper is clear and easy to follow, the technical novelty is limited, as the authors use the K-means to generate the pseudo-label for unsupervised learning. The authors can consider the recent proposed contrastive learning-based methods to further improve the performance. Moreover, the experiment results need to be further validated to remove the randomness of each experiment run.

**Paper Type:**

methodological development

**Questions To Address In The Rebuttal:**

The main concern is the experiment results in Table I. The authors need to ensure each method is fully trained and try to reduce the randomness of each experiment when comparing different methods.

**Special Issue:**

no

---

### Official Review · AnonReviewer1 · 2021-03-05

**Confidence:** 5
**Preliminary Rating:** 2
**Final Rating:** 2

**Summary:**

A multi-task learning framework for lesion segmentation using weakly supervised lesions is proposed. The method is evaluated on lesion segmentation from CT and dermoscopy images. The results show significant improvements in terms of dice value, averaged Hausdorff distance, and training time reduction.

**Strengths:**

Improvement of 6%, 13%, and 70% for dice metric, averaged Hausdorff distance (AVD), and training time respectively is reported.
Paper is mostly easy to read
Dermoscopy evaluation dataset is large
Multi-task learning results in improved segmentation performance compared to single-task learning

**Weaknesses:**

The network architectures (DeepLabV3+, ResNet101) for encoder and decoder are based on prior published work.

Multi-task learning, for joint classification and segmentation, has been investigated previously (see below). It is well known that joint learning improves specific task performance.

Yang X, Zeng Z, Yeo SY, Tan C, Tey HL, Su Y. A novel multi-task deep learning model for skin lesion segmentation and classification. arXiv preprint arXiv:1703.01025. 2017 Mar 3.

Amyar A, Modzelewski R, Li H, Ruan S. Multi-task deep learning based CT imaging analysis for COVID-19 pneumonia: Classification and segmentation. Computers in Biology and Medicine. 2020 Nov 1;126:104037.

There is no comparison against other multi-task learning methods or baseline SOTA segmentation methods.

Not all the CT dataset is used for evaluation.




**Deanonymize Review:**

no

**Detailed Comments:**

MAJOR: Was there a specific reason the authors did not use the full CT dataset? 1599 images are used instead of the full 32,735 computed tomography (CT) images? An evaluation study should be conducted using the full dataset or the reason for not doing so should be explained. If this is related to computational resources not being available the rest of the data can be used as testing.

Dataset size for the LiTS dataset should be mentioned and the authors should include if they have used all of this data or selected a subset again.

Have the authors considered developing a multi-modal (CT ad dermoscopy) multi-task segmentation method? Or multi-modal lesion segmentation method (multi-modal single task)?

Is the improvement in training time due to the use of minimal CT data?

What were the classification evaluation results (would be interesting to show as an appendix)?

MAJOR: The authors need to evaluation the segmentation accuracy against some state of the art segmentation methods such as Unet, Unet++, KiUnet. Comparison against their baseline segmentation method is not a fair comparison.


Minor comments:
Change “In this project” to “In this work”
Change “We train a model, denoted by a function f, such that for given an image” to “We train a model, denoted by a function f, such that for a given image”


**Final Rating Justification:**

More recent architectures should have also been investigated such as Unet++, Attention Unet, ResUnet, KiUnet. I am not expecting a full evaluation study for all the prior art but including one or two of these would make the work stronger. This is especially important and the technical novelty of the work is weak and the paper reads more of an application-oriented work

By looking at Table 7 the improvement of DeepLabV3+ is not significant vs Unet. A paired t-test should have been performed.

Also using 20% for validation and only 10% for testing seems odd. Why not use 20% for testing and 10% for validation.


**Justification Of The Preliminary Rating:**

Methodological novelty is very weak. Although the multi-task learning framework is shown to improve the quantitative results this is not new in the community. Multi-task learning has been investigated previously (see the references provided in the weakness section). The authors are also not making use of the full CT dataset and not comparing the results to other multi-taks learning methods or other SOTA segmentation results.

**Paper Type:**

validation/application paper

**Questions To Address In The Rebuttal:**

See above for MAJOR points.

**Special Issue:**

no

---

### Official Review · AnonReviewer2 · 2021-03-08

**Confidence:** 5
**Preliminary Rating:** 3
**Recommendation:** Poster
**Final Rating:** 3

**Summary:**

The authors present a multi-task learning method that jointly trains a model for voxel-wise lesion segmentation and lesion classification. Here, the lesion classification task classifies lesions into clusters based on weakly-labeled data in the form of RECIST measures (a readily available, but low-resolution annotation) of tumors. Experiments are performed on two modalities (CT imaging and dermatoscopic imaging) and demonstrate improvements over single-task, supervised training for segmentation.

**Strengths:**

•	Experimental evaluation is performed on two different modalities (CT images with lesions from two datasets and dermatoscopic skin lesions) tests applicability of the method to different datasets.
•	Good ablation study to test the optimal number of clusters (probably the most important hyperparameter) for the method.
•	Comparison to another state-of-the-art segmentation method (ACoseg) is provided, which provides good validation.
•	The multi-task learning paradigm reduces training times compared to the other state-of-the-art method.
•	The manuscript is very well written.


**Weaknesses:**


•	Input data appears to be limited to patches of imaging data focused on tumor/lesion regions of interest (ROI), which could potentially limit practical application of the method when ROI is unknown ahead of time.
•	Some training/experimental details, e.g. patch size, are not clear from the text.
•	Pixel-wise ground-truth segmentations are not provided for the DeepLesion dataset so an approximation using GrabCut is used as a. pseudo ground-truth (minor weakness since all three test algorithms used the same ground-truth).


**Deanonymize Review:**

no

**Detailed Comments:**

Sec. 4: It is unclear from the text how the imaging data is fed into the networks. It appears that the data consists of image patches that contain the tumors and lesions and therefore already assumes that the location of the object of interest is known. This is in contrast to the more general segmentation problem where the location of the object of interest is unknown. Please clarify if the test/validation/test sets are limited to image patches containing the tumors/lesions. If so, what are these patch parameters? How would the method perform on patches where no tumor exists?


Sec. 4.3: It would be very interesting to see what effect alpha in Eq 3 has on the segmentation results, especially since the chosen value alpha_seg has an order of magnitude (50 versus 1) more influence than alpha_cls on the loss (which suggests segmentation loss is very valuable).


Table 1: In addition to the mean values, it would be very interesting to report the standard deviation of the evaluation metrics in order to understand performance variation.


Sec. 4.3: What are the total number of trainable parameters for A1, ACoseg, and A1+L? This would be good to know to compare size of each model and put the results in context of these numbers.


Training time: How does the cluster classification loss (in model A1+L) affect training time compared to the segmentation-only network (A1)?


Minor comments:

Sec. 3: Please make sure to define variable used. While some are pretty clear, e.g. HxW is height and width, it would still be good to explicitly define these for readers to avoid any confusion. For example, “C” could be interpreted as the channel index in multi-channel 2D imaging, or it could be depth in a 3D volume (I know you are talking about 2D imaging here, but it’s good to be clear).


Sec. 4.3: “A1+L” is never defined in the first paragraph. From Sec. 5, it is clear that “A1+L” is A1 plus the cluster classification loss network, but this should be stated explicitly in Sec. 4.3.  Please clarify this in the text.


**Final Rating Justification:**

Thank you for your comments and revisions. Removing the question of lesion detection and focusing purely on lesion segmentation, I think the multi-task learning method to jointly train a model for voxel-wise lesion segmentation and lesion classification of weakly-labeled from clustering of RECIST values would be of interest to the MIDL community. Thank you for providing a link in the paper to source code for reproducibility.

**Justification Of The Preliminary Rating:**

The multi-task learning method to jointly train a model for voxel-wise lesion segmentation and lesion classification of weakly-labeled is interesting. The authors present a good set of experiments on different datasets to demonstrate improvements over single-task, supervised training for segmentation.

**Paper Type:**

methodological development

**Questions To Address In The Rebuttal:**

Please clarify how the imaging data is input to the network for training/validation/testing. Are image patches selected containing tumor/lesion ROIs? If so, this could potentially limit practical application of the method when ROI is unknown ahead of time.



**Special Issue:**

no

---

### Official Review · AnonReviewer4 · 2021-03-09

**Confidence:** 3
**Preliminary Rating:** 3
**Recommendation:** Poster

**Summary:**

1. The authors introduce the concept of multi-task learning to The authorsakly-supervised lesion segmentation, one of the most critical and challenging tasks in medical imaging.
2. The authors propose to jointly train a lesion segmentation model and a lesion classifier in a multi-task learning fashion, where the supervision of the latter is obtained by clustering the RECIST measurements of the lesions.

**Strengths:**

The authors evaluate their approach specifically on liver lesion segmentation and more generally on lesion segmentation in computed tomography (CT), as well as segmentation of skin lesions from dermatoscopic images. The paper is well-written and easy to follow. The Performance is good.

**Weaknesses:**

The authors show that the proposed joint training improves the quality of the lesion segmentation by 6% percent, while reducing the training time required by up to 70%. However, comparison study is somehow limited. How about the comparison with some recently published weakly supervised learning? (e.g., Hu, Shaoping, et al. "Weakly supervised deep learning for covid-19 infection detection and classification from ct images." IEEE Access 8 (2020): 118869-118883.) Conclusion is somehow not convincing.

**Deanonymize Review:**

no

**Detailed Comments:**

I would like to see more comparison studies results.
More parameter tuning details with ablation studies.

**Justification Of The Preliminary Rating:**

The paper is well-written and easy to follow. The Performance is good. However, comparison study is somehow limited. Conclusion is somehow not convincing. I think the authors can address these issues in their rebuttal.

**Paper Type:**

methodological development

**Questions To Address In The Rebuttal:**

I would like to see more comparison studies results.
More parameter tuning details with ablation studies.

**Special Issue:**

no

---

### Meta-Review · Area_Chair1 · 2021-03-24

**Recommendation:** Accept (Poster)

**Metareview:**

The authors have addressed all the points raised by the Reviewers to a different extent, including more experiments and discussion points to support the described method.  AnonReviewer1 indicated that a more extensive comparison with other recent and maybe more competitive baseline segmentation methods should have been included. I do agree that it would have been interesting to do so. However, all Reviewers agree that the paper represents a contribution as an application paper and, considering the rebuttal effort, I recommend to accept this work for publication.

**Paper Type:**

validation/application paper

---

### Decision · Program_Chairs · 2021-03-31

Accept